# Beyond Pellagra—Research Models and Strategies Addressing the Enduring Clinical Relevance of NAD Deficiency in Aging and Disease

**DOI:** 10.3390/cells12030500

**Published:** 2023-02-03

**Authors:** Morgan B. Feuz, Mirella L. Meyer-Ficca, Ralph G. Meyer

**Affiliations:** 1Department of Animal, Dairy, and Veterinary Sciences, College of Agriculture and Applied Sciences, Utah State University, Logan, UT 84322, USA; 2College of Veterinary Medicine, Utah State University, Logan, UT 84322, USA

**Keywords:** niacin, NAD, niacin deficiency, NAD animal models, nicotinamide, nicotinic acid, sirtuin, CD38, poly(ADP-ribose), aging

## Abstract

Research into the functions of nicotinamide adenine dinucleotide (NAD) has intensified in recent years due to the insight that abnormally low levels of NAD are involved in many human pathologies including metabolic disorders, neurodegeneration, reproductive dysfunction, cancer, and aging. Consequently, the development and validation of novel NAD-boosting strategies has been of central interest, along with the development of models that accurately represent the complexity of human NAD dynamics and deficiency levels. In this review, we discuss pioneering research and show how modern researchers have long since moved past believing that pellagra is the overt and most dramatic clinical presentation of NAD deficiency. The current research is centered on common human health conditions associated with moderate, but clinically relevant, NAD deficiency. In vitro and in vivo research models that have been developed specifically to study NAD deficiency are reviewed here, along with emerging strategies to increase the intracellular NAD concentrations.

## 1. History of NAD and Pellagra

Nicotinamide adenine dinucleotide (NAD) was discovered in the early 20th Century [1], and its redox capabilities and importance for cellular energy metabolism and ATP formation were recognized in the following decades [2,3]. The initial research focused on the roles of NAD^+^/NADH and NAD phosphate (NADP^+^/NADPH) as redox equivalents in animal, plant, and microbial metabolisms. Subsequent discoveries that NAD is also cleaved to form adenosine diphosphate (ADP)-ribose and is essential for protein poly- and mono-ADP-ribosylation, cyclic ADP-ribose, and NAD-dependent sirtuin activity significantly expanded the scientific knowledge of NAD functions [4,5,6,7,8]. Insights into the NAD functions that are unrelated to the classic redox reactions of NAD as a coenzyme have also contributed to the current understanding of pellagra, the most severe manifestation of NAD deficiency.

Pellagra, as the clinical disease caused by NAD deficiency, is characterized by a range of **D** symptoms: The first **D** stands for the eponymous clinical symptom of sun-sensitive **d**ermatitis, which led to the use of the Italian term “pelle agra” (meaning rough skin) as a basis for the name for the disease. Further manifestations are **d**ementia, and a relatively unspecific **d**iarrhea, which is associated with anorexia, which can exacerbate the causative problem of niacin deficiency. Untreated pellagra will ultimately lead to **d**eath.

Pellagra was first recognized as a clinical problem during the 18th Century in Spain and Italy, when epidemics of sun-induced skin lesions occurred among people working outdoors. During the late 19th Century, similar outbreaks began to occur in the southern United States. Pellagra remained a significant public health problem particularly in low-income parts of the population well into the middle of the 20th Century, when Joseph Goldberger’s and Conrad Elvehjem’s clinical trials and scientific studies pinpointed the lack of bioavailable dietary niacin in the corn-based diets of affected patients as the cause for pellagra [9,10]. That dietary nicotinic acid (niacin) or nicotinamide are essential for human NAD synthesis was not understood at that time, but after the discovery that food supplementation with these vitamin B3 molecules could prevent and cure clinical pellagra [11], flour products in the USA and Canada are routinely enriched with niacin to this day. The term vitamin B3 refers to a group of compounds that can all serve as NAD precursors, such as nicotinamide riboside (NR) and nicotinamide mononucleotide (NMN), in addition to nicotinic acid and nicotinamide.

Wheat flour fortification with niacin has been effective in preventing widespread clinical pellagra in the Americas and in other developed countries [12,13], and food-related pellagra is nowadays only occasionally diagnosed in developing countries undergoing food insecurity and famines. Despite the widespread success of supplementations to prevent clinical pellagra, there is still evidence of NAD deficiency even in developed countries, which can be detected by a low NAD/NADP ratio during a blood analysis [14], which has significant health implications, as outlined in the following sections.

## 2. NAD Deficiency Problems in the Modern Day

As mentioned above, besides the importance of the NAD^+^/NADH pair for cellular redox reactions, NAD, specifically its oxidized form NAD^+^, is also an essential cofactor for numerous non-redox enzyme reactions. A number of NAD^+^-dependent enzymes degrade NAD^+^ in order to perform important regulatory and signaling functions [15,16]. For example, members of the poly(ADP-ribose) polymerase family (PARPs or ARTDs) are major NAD^+^-cleaving enzymes that are involved in DNA repair and a plethora of other cellular functions (these are reviewed in [5]). The NADases CD38, CD157, and SARM1 (sterile alpha and toll interleukin receptor motif-containing protein 1) are ecto-enzymes, with ADP-ribosyl cyclase activity being involved in immunoregulation, together with the ecto-nucleotidase CD73. Furthermore, the protein deacylase enzymes in the sirtuin family consume NAD to perform complex functions that regulate cellular metabolism. The activity of those enzymes causes ongoing tissue NAD cleavage, which can lower the tissue NAD levels if NAD generation is inadequate to counteract its consumption.

Importantly, if the tissue NAD levels decline, the available oxidized NAD^+^ generally decreases disproportionally to the reduction of NADH, which leads to a shift in the overall NAD^+^/NADH redox potential, in addition to a general NAD^+^ paucity. This combined effect is therefore expected to cause a shift in both the activity of NAD-dependent redox enzymes and the activity of NAD^+^-consuming enzyme groups.

In recent years, we have seen accumulating evidence that decreasing NAD levels are associated with multiple disease states and health problems ranging from metabolic dysfunction, neurodegeneration, and inflammatory diseases to problems with fertility and embryo development [17,18,19,20,21,22,23,24,25,26,27,28,29,30,31,32,33,34,35]. Furthermore, it has been shown that a marked drop in the NAD levels also occurs during normal aging, where it might contribute to various aspects of the aging-related health decline [36], and it is becoming increasingly clear that NAD^+^ homeostasis is closely connected to health span and lifespan [37]. As a result, there has recently been intense scientific interest in the potential benefits provided by NAD-boosting supplements. The number of animal studies and clinical trials exploring the benefits of such improved NAD availability in the context of numerous diseases has steadily increased over the last decade. While many studies have found certain health benefits from NAD-boosting therapies, the underlying molecular and cellular mechanisms are often not well understood. The renewed interest in NAD research has fueled an ever-increasing need for research models that permit us to find new insights into the dynamics of NAD metabolism at the cellular and organismic levels, and ultimately, allow us to understand the connection between alterations to the NAD metabolism and disease etiology, which is the focus of this review.

### 2.1. NAD and Aging

Although the underlying reasons are not well understood, tissue NAD content decreases with age in both humans and rodents [17,38,39]. This NAD decline is thought to result from an imbalance between NAD biosynthesis and NAD consumption [16,40]. Increasing age is associated with a lower activity level of NAMPT, which is the rate-limiting enzyme in the salvage pathway that generates most of the NAD content in mammals [40,41]. Recent studies have also identified an increase in activity of the NAD hydrolase CD38 located on the cell surface of pro-inflammatory macrophages as a major contributor to aging-associated NAD decline [42,43,44].

Interestingly, tissues from older individuals exhibit more DNA damage, which may further exacerbate NAD depletion due to the DNA strand break-dependent activation of PARP enzymes to form poly(ADP-ribose) (PAR) from NAD [45,46]. In this context, it is important to note that the observation that the maximum capacity of PAR formation positively correlates with the maximum lifespan of different species [47,48]. This points towards the potential benefits of high PARP activity levels for maintaining genetic and genomic stability, which is conducive to longevity. Similar to PARP enzymes, sirtuins have also been implicated in the regulation of longevity and are regarded to be regulators of aging [49]. The epigenetic modifications mediated by some sirtuins provide transcriptional control, which regulates the energy metabolism pathways and mitochondrial activity, again linking NAD availability to aging [50].

Increased inflammation, genomic instability, epigenetic alterations, and mitochondrial dysfunction are all considered to be hallmarks of aging [51]. This raises the key question whether the age-associated NAD decline causes the health decline that typically occurs with increasing age, or if the aging-related health decline, i.e., some of the “hallmarks of aging”, result in lower NAD levels as a consequence of the aging process. The answer may be that the relationship between aging and NAD decline is interdependent—as though it is in a vicious cycle—and therefore, it is more complicated, which is probably the most likely scenario.

### 2.2. NAD and Skin Photoaging

In line with photosensitive dermatitis as a major pellagra symptom, fish and mice experimentally maintained under niacin-restricted conditions with resulting mild NAD deficiencies have an increased skin sensitivity to UV radiation and an increased cancer incidence [52,53]. Human skin cells cultured under niacin-deficient conditions shift their NAD redox potential more towards reduced NADH levels, and simultaneously, they accumulate reactive oxygen species (ROS). They become also more sensitive to UV radiation, similarly to what has been described for the in vivo study of animals [54,55]. It is therefore plausible that the age-related NAD decline and shift in the NAD/NADH redox potential contributes to increased photosensitivity in aging human skin and predisposes aged skin to carcinogenesis.

### 2.3. NAD and Cancer

One potential and widely discussed connection between the aging-associated NAD decline and aging-associated pathologies is the involvement of suboptimal NAD levels in cancer development [36]. DNA damage in the form of DNA single- and double-strand breaks activates the nuclear NAD-consuming enzymes PARP1, PARP2, and PARP3 to produce PAR, which is required for efficient DNA base excision repair and the repair of DNA double strand breaks via non-homologous end joining, but not homologous recombination repair. The synthesis of PAR requires NAD as substrate, where large amounts of NAD are cleaved into nicotinamide (NAM) and ADP-ribose, which attach to target proteins as a posttranslational modification that is important in DNA damage repair signaling pathways [56]. In humans, the PARP protein family comprises 16 enzymes with known PAR polymerase or mono(ADP-ribose) transferase activity, which makes them major consumers of NAD. PARP1, PARP2, PARP3, and tankyrase are the main PARP family members that recognize DNA damage and form PAR to facilitate efficient DNA repair, and thereby, maintain genomic stability [57,58,59]. Excessive PAR formation can also result in cancer-preventing apoptosis in highly damaged cells [60]. It is therefore plausible that insufficient NAD availability could lead to an impaired ability of cells with DNA damage to undergo apoptosis, resulting in an accumulation of damaged cells with oncogenic potential. The important function of PARPs in DNA repair has been exploited in cancer therapies which rely on the concept of synthetic lethality. In these therapies, certain cancers that are inherently deficient in homologous DNA repair pathways are treated with potent PARP inhibitors, which sensitize them to a treatment with cytotoxic agents that induce DNA strand breaks. Most famously, such cancers include BRCA-deficient breast cancers, but PARP inhibitors have more recently also been used to treat ovarian, pancreatic, and prostate cancers [61].

The protection of DNA integrity through the activity of PARP enzymes suggests that maintaining NAD availability at appropriately high levels could be beneficial for cancer prevention. Indeed, various lines of evidence obtained from animal models and human epidemiological trials indicate that the insufficient availability of NAD will increase the cancer risk [62,63]. Consistent with these data, various studies point towards a benefit of maintaining sufficiently high NAD levels, such as through supplementation with NAD precursors for the prevention of cancer [64].

Based on these insights into the prevention of carcinogenesis by maintaining adequate levels of NAD, therapeutic concepts have emerged recently that aim to destroy existing cancer cells by inhibiting the NAD metabolism and synthesis pathways. The underlying rationale for these strategies is the observation that tumor cells have high NAD demands due to a shift of the metabolic preferences to anaerobic glycolysis, even under normal oxygen conditions. This is known as the Warburg effect. Therefore, the inhibition of NAD synthesis pathways, e.g., by blocking the rate-limiting enzymes such as nicotinamide phosphoribosyltransferase (NAMPT) in the salvage pathway or nicotinic acid ribosyl transferase (NAPRT) in the Preiss-Handler pathway (Figure 1) is currently being explored as a potential component of cancer therapies [18,65,66].

### 2.4. Iatrogenic and Nutritional NAD Deficiency

Cancer is also an underlying cause of clinical or subclinical NAD deficiency, as observed regularly in industrialized countries. Cancer patients can experience low NAD levels, as the malignant changes can directly interfere with NAD synthesis, as is the case in carcinoid cancer patients [67]. In addition, cancer therapies and the associated chemotherapy drugs can secondarily deplete the amount of NAD by causing DNA strand breaks by PARP activation, which can in some cases even result in clinical pellagra-like symptoms [68,69].

Iatrogenic NAD deficiency can also occur following the use of a drug treatment for other conditions than cancer, such as tuberculosis and HIV [70,71,72,73,74]. Isoniazid (INH), a drug used for tuberculosis prevention, forms adducts with NAD and NADP and prevents the absorption of dietary niacin. This ultimately increases the risk for NAD deficiency and even clinical pellagra, in particular, in combination with additional risk factors such as HIV infection, a lack of access to quality food, a low body weight, alcohol overconsumption or, in females, lactation [75,76]. The drug 5-fluorouracil, which is used either in antiviral therapies or as part of chemotherapy regimens for the treatment of gastrointestinal malignancies, induces pellagra symptoms in patients, which is likely due to impaired tryptophan metabolism and insufficient uptake of dietary NAD precursors [69].

While nutritional pellagra due to food insecurity is uncommon in developed countries, it still occurs there in the context of anorexia nervosa [19,20,21], alcoholism [22,23], and in malabsorptive disorders such as Crohn’s disease [24] or Hartnup’s disease, and during end-stage renal disease [25]. A specialized metabolic situation occurs during pregnancy, during which the niacin requirements and metabolism change and more tryptophan conversion to NAD is observed [26], indicating that tryptophan supplementation might protect pregnant females and their offspring from potential risk of NAD deficiency [27].

### 2.5. NAD in Reproduction and Development

NAD^+^/NADH and NADP^+^/NADPH are essential redox equivalents for a plethora of metabolic reactions, including the successive reactions in steroid hormone synthesis. Another important group of metabolic reactions that needs sufficient NAD and NADP redox equivalents are the metabolic pathways involved in oxidative stress response, such as glutathione synthesis. The cellular defense against redox stress and potential oxidative damage is especially important for fertility and reproduction, and a diminished oxidative stress defense causes a loss of gamete quality and zygote developmental potential [28,29].

With an increasing maternal age and declining ovulation rates, the NAD content in oocytes drops, and simultaneously, the oocyte quality deteriorates due to increased ROS and DNA damage, resulting in reduced fertilization rates and a lower developmental potential of zygotes. These adverse effects of age-associated NAD decline can be ameliorated by supplementation with NAD precursors such as NR or NMN (Figure 1) [30,31,77]. Several in vitro studies have corroborated the risk of increased oxidative stress for follicular development and oocyte quality. They underline the importance of adequate NAD levels, in particular, for in vitro oocyte cultures and embryo competency in assisted reproductive techniques [78,79,80,81].

In males, the age-associated NAD decline aligns well with an age-associated reduction in testosterone, as well as a decreased testicular weight. In a mouse model, in which the tissue NAD levels in young males were reduced by dietary intervention, decreased tissue NAD levels caused progressive testicular hypotrophy and a cessation of sperm production, which is similar to the situation in older males [32].

The crucial importance of appropriate NAD levels for healthy human embryonic development is evident in families with genetic loss-of-function mutations in gene-encoding enzymes involved in the kynurenine (de novo) pathway, where the resulting disruption of NAD synthesis from tryptophan lowered the levels of circulating NAD. Both the humans affected by the original mutations and the mice, in which the mutations had been recreated, suffered from congenital vertebral and heart malformations which in the experimental mouse model, could have been prevented by niacin supplementation during gestation [33]. Many known teratogens affect NAD levels, for example, by changing its metabolism functions, by decreasing its intracellular availability, or by decreasing its production rate. Such teratogens may therefore exacerbate the problem of suboptimal NAD levels caused by inadequate niacin intake or other health problems, giving rise to plausible mechanisms for the origination of many birth defects. This theory is currently supported by the results of a large meta-analysis [82,83].

### 2.6. The Role of NAD in Degenerative Diseases

Aging has long been known as a major risk factor for many degenerative diseases, most prominently for neurodegenerative diseases such as Alzheimer’s disease (AD) or Parkinson’s disease (PD) [84]. Emerging findings emphasize the importance of NAD for neuronal resilience, pointing, in particular, to the role of NAD as an essential metabolite in the cellular bioenergetic processes that are at the core of synaptic plasticity and neuronal stress resistance. Based on the insight that brain NAD concentrations decline with increasing age [17], the view that healthy NAD levels are important to counteract the degenerative processes that underlie neurodegenerative diseases such as AD, PD, Huntington’s disease, and amyotrophic lateral sclerosis has emerged [34]. Further support for this theory stems from observations of decreased NAD levels in various degenerative diseases, including neurodegenerative AD and PD, as well as cardiovascular disease and muscular dystrophy [35], and from the observed benefits of supplementation with NAD precursors. One example is the health benefits that have been seen for nicotinic acid supplementation in an AD mouse model [85].

Many other degenerative diseases seem to be connected to imbalances in NAD homeostasis as well, and consequently, they might be amenable to NAD-boosting interventions. For example, an NMNAT-1 mutation that affects cellular NAD synthesis (Figure 1) causes a severe form of early-onset inherited retinal dystrophy [86]. Similarly, NAM supplementation can ameliorate age-related macular degeneration [87]. The potential therapeutic benefits of NAD supplementation are also being discussed for age-related chronic glaucoma [88], where NAD-boosting strategies in animal models were able to successfully prevent or intervene the progression of the disease [89]. Furthermore, recent experimental evidence has linked decreasing NAD levels in osteoblast progenitor cells with a loss of bone mass during aging, which is a potential factor in the etiology of osteoporosis [90]. Moreover, age-related sarcopenia in human skeletal muscles is characterized by mitochondrial dysfunction with reduced NAD levels due to perturbed NAD synthesis [91]. In another example, cardiac muscle cells experience NAD level decline during atrial fibrillation, while supplementation with the NAD precursor NMN can prevent heart failure through maintaining the tissue NAD levels and mitochondrial homeostasis [92,93]. Finally, multiple organ fibrosis is associated with disrupted NAD homeostasis and can be ameliorated by strategies that increase the NAD levels [94].

### 2.7. NAD and Infectious Diseases

NAD metabolism in host organisms is emerging as an attractive target in the fight against microbial and viral infections, and evidence for the importance of appropriate NAD levels and the potential benefit of supplementation is accumulating [95]. Interestingly, infectious diseases can directly interfere with NAD production, and thus cause intracellular deficiency. Infection with the Human Immunodeficiency Virus (HIV) alone and co-infections of HIV and hepatitis C can cause a significant decline in the intracellular and tissue NAD levels, likely through the oxidative-stress-induced overconsumption of NAD which might cause pellagra-like clinical symptoms in some patients. This has led to the hypothesis that niacin supplementation might be useful as an AIDS preventative [96,97,98,99,100].

Other viral infections, for example, with murine coronavirus, upregulate the expression of pro-viral PARP enzymes within their cytokine response [101]. More recently, the disproportionately more frequent adverse outcomes due to COVID-19 infections in people that are older, obese, or have type 2 diabetes has led to suggestions that lower-than-normal NAD levels, which are typical of these groups [18], are a risk factor for a more severe course of the disease [102]. Several PARP enzymes (PARP9 and PARPs 11-14) have antiviral activities [103]. SARS-CoV2 infection causes an upregulation of PARPs that consume NAD during their antiviral response [104]. The antiviral immune defense and inflammatory reactions induced by SARS-CoV-2 infection thereby dramatically deplete the NAD levels, in particular, during a cytokine storm, when the increase in the amount of ROS produced by the antiviral response causes significant oxidative stress. This can lead to the excessive depletion of cellular antioxidants such as glutathione, resulting in more DNA damage, which activates the PARP enzymes, and thus depletes the NAD content further. Cytokine production is normally modulated to an appropriate degree though the activity of NAD-dependent sirtuin enzymes. Excessive NAD consumption depletes mitochondrial NAD stores, leading to insufficient sirtuin activity, and thus, it results in an unchecked cytokine response [105,106].

In order to better understand the molecular and cellular changes that a loss of NAD homeostasis can cause and how NAD decline and the discussed diseases are connected, the need for cell and animal models that can faithfully model the NAD changes seen in human diseases has become more urgent.

## 3. Models of NAD Deficiency

### 3.1. In Vitro Models

NAD-deficient cell culture models can be readily achieved by removing any NAD precursors, such as niacin, NAM, and NR, from the media or supplements. The early work showed that when they were cultured in niacin-deficient media in combination with DNA-damaging chemicals such as dimethyl sulfate (DMS), the cells displayed a marked increase in the number of DNA strand breaks, lower PAR levels, and a decline in cell viability [107,108]. When the NAD content was depleted to 80–90% of that of the controls, the niacin-deficient cells still grew at the same rate, suggesting that cell growth is not heavily impacted by NAD deficiency when they are in a culture. This phenomenon is not reflected when one is using in vivo models, even during severe instances of niacin deficiency [109]. Thus, the extrapolation of the results from the cultured cells to whole animals is difficult. An additional limitation is that most of the cultured cells do not accurately reflect all of the in vivo pathways for NAD production from niacin since only hepatocytes and kidney cells have been shown to possess the ability to synthesize NAD from tryptophan via in the kynurenine (de novo) pathway [54,110]. Nonetheless, successful niacin-deficient cell culture models for carcinogenesis have been reported [54,55,111]. Benavente et al. identified NAD-dependent pathways using cultured skin cells, which demonstrated an accumulation of ROS, DNA damage, and apoptosis due to the increased expression and activity of NADPH oxidase when it was cultured in niacin-deficient media [54,55]. This model also revealed that culturing in niacin-deficient media causes the upregulation of sirtuin expression and decrease PARP activation following photodamage, which is reversible with niacin repletion [111].

### 3.2. In Vivo Models

In order to study body-wide NAD deficiency and the pathologies associated with suboptimal NAD levels, is seems essential that animal models used for such research can actually be NAD-deficient ones. Otherwise, such research would investigate the effects of NAD supplementation above the normal levels, instead of modeling a condition associated with NAD deficiency. Unlike humans, commonly used laboratory animals are well protected from becoming NAD deficient, and these natural defenses must be overcome to generate animal models that are suitable for investigating the role of suboptimal NAD in pathogenesis, as well as the benefits of the supplementations.

#### 3.2.1. Historical Models

The first experimental phenotype closely resembling the symptoms of pellagra in humans was identified using a dog model [112]. In 1937, black tongue disease, the manifestation of pellagra in dogs, was cured using nicotinic acid that was isolated from liver extracts [113,114]. Additional early nutritional studies on laboratory animals reported the requirement of dietary niacin for survival and the adequate growth of rats [115], mice [116], guinea pigs [117], and rabbits [118]. Mice and rats are the most commonly used laboratory animals due to their small size, mild temperaments, and short reproductive cycles and lifespans [119]. Additionally, mice are amenable to genetic manipulations such as adding or deleting genes, and they are known as transgenic or knockout animals, respectively, and breeding studies following such genetic manipulation, making them the currently most popular animal models in biomedical research [119]. A rodent model is therefore ideal for studying niacin deficiency in a laboratory setting and, unsurprisingly, the majority of the published animal-based niacin deficiency experiments have utilized mice or rats.

However, a major challenge in developing a dietary niacin-deficient animal model is that NAD can by synthesized from tryptophan in mammals, a reaction that occurs via the kynurenine (de novo) pathway [120,121,122] (Figure 2). Tryptophan is an essential amino acid that is used in protein biosynthesis, but it is also essential for the formation of pyridine nucleotides and serves as the precursor for the neurotransmitter serotonin, and therefore, it cannot be eliminated from the diet. Additionally, the conversion ratio of tryptophan to NAD varies markedly between the species. Humans cannot make sufficient NAD from tryptophan alone since they have a poor conversion ratio, which has been experimentally determined to be 1/60-70 [121,123], which is different to that of 1/33 in rats. Mice are even more efficient at converting tryptophan into NAD than rats are, and they are able to completely fulfill their NAD needs from tryptophan and will not become NAD deficient even if their diet lacks any vitamin B3 [53,120]. The initial studies on the niacin nutritional requirements of guinea pigs suggested that they had limited tryptophan-to-niacin conversion capabilities, leading researchers to propose that guinea pigs may more accurately represent the human metabolism [117,120]. However, the follow-up work revealed that niacin-deficient diets using casein with low levels of tryptophan as a protein source resulted in only modest, transient changes in the blood and bone marrow NAD concentrations in guinea pigs, leading to higher rates of mortality compared to those of rats [120]. This suggests that guinea pigs are not an ideal model for severe niacin deficiency, and their usability may be limited to modelling transient and mild NAD loss.

#### 3.2.2. Dietary Rat Model with Transient NAD Deficiency

The first NAD-deficient rat models revealed that a niacin-free diet containing low concentrations of tryptophan can affect the NAD concentrations and PARP activity of the animals, which was hypothesized to signal an increased cellular susceptibility to DNA damage [109,124]. However, it was also reported that maintaining rats in an NAD-deficient state was difficult, and it depended not only on the concentrations of dietary niacin and tryptophan, but also on the age of the animal, and as animals became adults, they became resistant to the NAD deficiency induced by a lack of dietary niacin [109,124]. It has been postulated that following a prolonged niacin-free diet period, the animals may be able to convert tryptophan into NAD more efficiently, resulting in a recovery from the niacin deficiency symptoms [120]. This transient nature of NAD deficiency, as well as the fact that it could only be achieved in prepubertal juveniles, suggests that rats may not be an ideal model for studying the effects of NAD deficiency on reproduction or the aging process.

Regardless, the juvenile rat model has been used successfully in several breakthrough studies examining the effects of NAD deficiency from the perspective of the effects of a chemotherapy treatment [125,126,127,128,129,130]. Chemotherapy agents target rapidly dividing cells, which means that they are effective against tumor cells, but they also damage the cells in the hair follicle, gastrointestinal tract, and bone marrow [125]. Long-term or aggressive chemotherapy treatments can cause the formation of secondary malignancies from excessive DNA damage. Cancer patients undergoing chemotherapy are often malnourished and can display NAD deficiency with pellagra-like symptoms. NAD deficiency through niacin-free diets combined with a chemotherapy treatment in rats was shown to exacerbate bone marrow suppression and oxidative stress, ultimately leading to the development of secondary cancers [125,126,128,131]. In addition, dietary supplementation with niacin lengthened the latency of secondary cancer development following a chemotherapy treatment, while also increasing the bone marrow concentrations of NAD and PAR, suggesting that niacin supplementation may be beneficial for patients undergoing chemotherapy with a compromised nutritional status [129,130].

#### 3.2.3. Dietary Mouse Models with Mild NAD Deficiency

Mouse models of mild NAD deficiency have been reported using diets containing low, but adequate levels of tryptophan in combination with niacin depletion [132,133]. Gut microbiota have been postulated to be a niacin source, with a contributing role in pellagra-induced nausea [133]. The analysis of the microbiomes of mice fed a low-niacin diet revealed changes in the fatty acid metabolism and amino acid metabolites, suggesting that there may be therapeutic potential of the microbiota in treating patients with pellagra [133]. Differences in the expression of Ppar, a master regulator of adipogenesis in epididymal white adipose tissue, as well as altered metabolic profiles and changes in insulin sensitivity, were observed in animals fed diets lacking dietary niacin compared to those of the animals fed NR-supplemented diets [132,134]. Interestingly, female mice kept under the same conditions showed no differences between the niacin-deficient and control groups, suggesting that females are more resistant to the effects of dietary niacin depletion than males are [135]. However, dietary niacin restriction combined with genetic mutations in the kynurenine pathway were shown to decrease the NAD levels during pregnancy, and this resulted in embryo loss and congenital malformations [136].

#### 3.2.4. Pharmacological Animal Models

Pharmacological agents have been used to overcome the limitations of rodent models for modeling human-like niacin deficiency, which is described above, to induce NAD-deficient phenotypes [137,138]. However, a major limitation of using pharmacologic agents is the potential for increased drug-associated toxicities or off-target effects. Photosensitive dermatitis, likely caused by prostanoids and ROS production, is a hallmark abnormality of pellagra-associated NAD deficiency [137]. A combination of a niacin-deficient diet and the administration of 6-aminonicotinamide (6-AN), which inhibits the metabolism of tryptophan to NAD, resulted in a manifestation of photosensitivity dermatitis following an ultraviolet B (UVB) treatment, which is similar to that which was seen in humans with pellagra [137], but it also caused central nervous system toxicity [139]. Anti-tuberculosis agents, such as INH, cause gastric symptoms and nausea, which may be indicators of pellagra [138]. Mice fed a niacin-deficient diet with a subsequent INH treatment developed pica, which is regarded to mimic human vomiting, a symptom that is indicative of pellagra. This mouse model of pellagra-related nausea could potentially be used to determine how patients taking drugs that cause secondary pellagra can be treated better [138].

#### 3.2.5. Genetic Mouse Models

In mammals, tryptophan can be converted into NAD through the kynurenine (de novo) pathway (Figure 2). Therefore, completely NAD-deficient animal models cannot be generated using dietary manipulation alone. Terakata et al. generated the first truly NAD-deficient mouse model by removing the quinolinic acid phosphoribosyltransferase (*Qprt*) gene [140,141] (Figure 2). In the kynurenine pathway, alpha-amino-beta-carboxy-muconate-semialdehyde (ACMS) is either enzymatically converted into alpha-amino-muconate-semi-aldehyde (AMS) to form picolinic acid and acetyl coenzyme A (acetyl-CoA) by ACMS decarboxylase (ACMSD) or non-enzymatically to quinolinate, a precursor for NAD synthesis. Feeding a niacin-free diet to *Qprt^−/−^* mice resulted in significantly decreased NAD levels. A limitation of this system is the accumulation of quinolinate, a known neurotoxin and agonist of N-methyl-D-aspartic acid receptors [121,140,141]. An alternative genetic knockout mouse model, again focused on the manipulation of the kynurenine pathway, removed the *Tdo* gene coding for the enzyme L-Trp-2,3-dioxygenase (TDO), which along with indoleamine-2,3-dioxygenase (IDO), provides the first and rate-limiting step of catalyzing the oxidative cleavage of tryptophan for it to become N-formylkynurenine. However, *Tdo^−/−^* mice fed niacin-deficient diets did not develop the characteristic NAD-deficient phenotype, suggesting that a sufficient amount of NAD was still being synthesized from tryptophan in the absence of TDO by IDO alone [141,142,143].

An alternative approach to deleting genes important in the kynurenine pathway uses transgenic mice with the tetracycline-inducible expression of human *ACMSD* in order to deplete ACMSD, and thereby, develop an NAD-deficient mouse model. Palzer et al. reported that such mice, which are characterized by an acquired niacin dependency (“ANDY” mice), developed mild-to-severe NAD deficiency when they were fed a niacin-free diet [122]. In the ANDY mouse model, chronic NAD deficiency can be achieved, which is necessary for investigating reproductive and age-related pathologies. For instance, young adult ANDY mice on a niacin-deficient diet developed low NAD levels that were similar to those of chronologically old males undergoing natural age-related NAD decline, which resulted in spermatogenic failure [32]. Although the ANDY mouse has been shown to be a versatile tool for investigating the effects associated with niacin-deficiency using an animal model, a potential drawback of the system is that doxycycline (dox) must be administered in the drinking water to induce the expression of the ACSMD transgene, which necessitates the careful use of appropriate control animals [122].

Another strategy to lower NAD targets the salvage pathway, which creates NAD from NAM. The deletion of the gene that encodes the enzyme nicotinamide phosphoribosyltransferase (NAMPT, previously also known as Visfatin or PBEF (Pre-B-Cell Colony Enhancing Factor 1)) successfully prevents creation of NAD from NAM in the salvage pathway (Figure 2). The *NAMPT* gene is ubiquitously expressed, albeit with varied tissue-dependent levels, reflecting the varying degree of NAD biosynthesis among the different tissues [144,145]. A complete somatic knock-out of the *Nampt* gene is not compatible with life, i.e., a knock-out of *Nampt* during early development is embryonically lethal, and the inactivation of the gene in adult animals causes death within 5-10 days [146]. Tissue-specific *Nampt* knock-out strategies based on floxed alleles have been employed to demonstrate the importance of NAMPT activity for T and B cell development [147] for the glucose-stimulated insulin release in pancreatic β cells [145], for the viability of Schwann cells in the peripheral nervous system [148], and to maintain mitochondrial homeostasis in the hippocampal neurons [149].

The genetic deletion of *Naprt*, *Nadsyn*, *Nrk1*, *Nrk2* and *Nmnat3* did not result in lowered NAD levels, which is presumably due to the diversity of Vitamin B3 precursors of NAD synthesis and the partial redundancy of the associated biosynthetic pathways [150,151,152]. *Nmnat1* and *Nmnat2* deletion in a homozygous state was embryonically lethal, and *Nmnat1* heterozygous animals did not have lowered NAD levels [153,154].

**Figure 2 cells-12-00500-f002:**
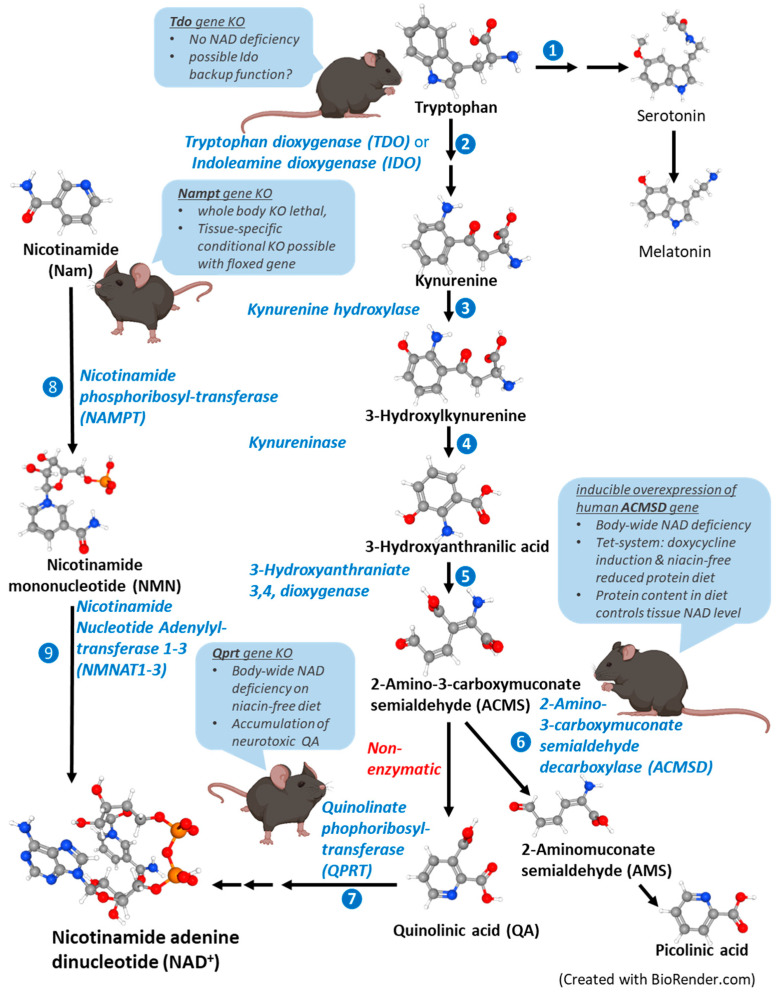
Genetically modified mouse models of altered NAD biosynthetic pathways. Tryptophan can be converted into nicotinamide adenine dinucleotide (NAD) by enzymes of the kynurenine pathway. (1) Tryptophan is an essential amino acid and, among other things, it is required for serotonin and melatonin synthesis; (2) the enzymes tryptophan dioxygenase (TDO) and indoleamine dioxygenase (IDO) convert tryptophan into formyl-kynurenine; (3–5) the enzymes kynurenine hydroxylase, kynureninase and 3-hydroxyanthraniate 3,4, dioxygenase convert kynurenine into the intermediate 2-amino-3-carboxymuconate semialdehyde (ACMS) in successive steps; (6) ACMSD can be converted to 2-aminomuconate semialdehyde (AMS) by the enzyme 2-amino-3-carboxymuconate semialdehyde decarboxylase (ACMSD) or rearranges non-enzymatically to become quinolinic acid (QA); (7) QA serves as substrate for quinolinate phophoribosyl transferase (QPRT) and it undergoes successive reactions that lead to NAD production. In order to prevent effective tryptophan-to-NAD conversion and to generate animal models of NAD deficiency, the following genetically modified mice were produced: one with a gene knock-out of *Tdo* to prevent step 2 [141,142,143], one with a gene knock-out of *Qprt* to prevent step 7 [140,141], and one with enhanced ACMSD activity that depleted ACMS in step 6 through an inducible *ACMSD* transgene [122]. In the salvage pathway (left), the enzyme nicotinamide phosphoribosyltransferase (NAMPT) uses nicotinamide (Nam) to generate nicotinamide mononucleotide (NMN) (step 8), which is further processed until it becomes NAD by nicotinamide nucleotide adenylyltransferase enzymes 1-3 (NMNATs 1-3) (step 9). Gene knock-out of the *Nampt* gene prevents NAD synthesis in the salvage pathway. While the complete body-wide loss of the *Nampt* gene is embryonically lethal, tissue-specific gene ablations are used to study the effects of local NAD depletion [145,146,147,148,149]. The chemical structures were retrieved from PubChem (https://pubchem.ncbi.nlm.nih.gov/, accessed on 18 January 2023).

## 4. Modeling NAD Turnover (“Flux”)

With the advent of technological advances in metabolite measurement technologies, such as nuclear magnetic resonance (NMR) and mass spectrometry (MS), metabolomics investigations in conjunction with metabolic flux are greatly improving metabolic research [155]. However, a limitation of modeling metabolic flux is that fluxes cannot be directly measured and must be inferred, for example, though the use of isotope traces, followed by integration using mathematical computation. Therefore, substantial pathway information, including compartmentalization, must first be obtained prior to flux quantitation at the system level [155]. Early work tracing NAD biosynthesis in vivo utilized ^14^C pulse-labeling techniques in mice, which revealed that the administration dose determined whether nicotinic acid or NAM was the more efficient precursor of liver NAD synthesis [156]. Additionally, ^1^H NMR spectroscopy has also been successfully used to detect and quantify NAD and its corresponding intermediates in human HEK293 and blood cells [157]. Recent isotope tracer methods, combined with mathematic modeling, have been used to quantitate NAD fluxes in both in vitro and in vivo models, where NAM was reported to be the major NAD source [110]. Notable variations in NAD flux were observed in many mouse tissues, with the liver actively synthesizing NAD from tryptophan and orally administered NR. Alternatively, NAD synthesis in tissues other than the liver depends on the circulating NAM. In contrast, cultured cells can readily use both NR and NAM in the production of NAD [110]. Isotope tracing and MS has also been recently utilized to evaluate age-related changes in NAD metabolism, since it is well documented that NAD declines with age [40,158,159]. Interestingly, the tissues of aged (25-month-old ones) mice had only a ~30% NAD depletion rate, and they were unaffected by the concentrations of circulating NAD precursors and the unimpaired synthesis of NAD from tryptophan compared to the young (3-month-old ones) mice. These results suggest that age-related NAD decline is likely due to increased NAD consumption, indicating potential implications for NAD supplementation [159].

## 5. Current Strategies to Increase NAD

Declining levels of blood and tissue NAD levels are a hallmark of aging [51,160]. The upregulation of NAD levels has therefore been investigated in a plethora of animal and clinical investigations of strategies for the potential treatment of a variety of age-related diseases and other pathological conditions, including AD, dementia, hyperphosphatemia, hypertension, obesity, PD, chronic fatigue syndrome, photoaging of skin, psoriasis, skin cancers, type 1 and type 2 diabetes, and schizophrenia [161]. Two major strategies for the upregulation of NAD levels have been employed.

Firstly, the inhibition of NAD-consuming enzymes has been suggested to elevate the NAD levels, for example, by inhibiting the NAD hydrolase CD38. The activity of this enzyme, which is mainly an ecto-enzyme in circulating immune cells, and which also has cyclic ADP-ribose synthase activity, increases with age and accounts for much of the NAD decline in the aging individual [43]. Flavonoids such as apigenin [162] inhibit CD38, and a potent and selective thiazoloquin(az)olin(on)e CD38 inhibitor, 78c, has been shown to restore low NAD levels in mouse models of aging [163]. Other proposed targets have included enzymes of the PARP family (also known as ADP-ribosyltransferases, ARTD, [5]). However, the inhibition of PARP1 and PARP2 (ARTD1 and ARTD2), which account for the majority of the NAD consumed by enzymes of this family, for the purpose of raising the NAD levels in healthy individuals has been controversial, as they have important roles in DNA repair and DNA damage repair signaling.

Secondly, supplementation with dietary precursors of has been used to increase low levels of NAD. As it is explained elsewhere in this review, nicotinic acid (NA) was one of the first compounds used for the treatment and prevention of pellagra. A decrease in the NAD levels can be prevented or corrected by the administration of other dietary precursors of NAD synthesis, such as NAM, NMN, or NR (Figure 1). The oral bioavailability of such precursor molecules has been subject to extensive research, as these precursors, in part, undergo conversion by the gastrointestinal microbiome before being absorbed into the portal vein as NR, NAM, NA, or NMN. These compounds are distributed throughout the body and cross cell membranes either passively (NAM) or by facilitated transport. NR and nicotinic acid riboside (NAR) transport into the cell has been reported to be mediated by NRT1 in yeast [164] and by equilibrative nucleoside transporters ENT1, ENT2, and ENT4 [165], while NA is transported by several different proteins, including organic anion transporter 2 (OAT2 and SLC22A7) [166]. It has been proposed that NMN, which is a nucleotide, and as such, is not typically expected to be able to cross the cell membrane, may be able to enter cells by means of the transporter SLC12A8, but this has been controversially discussed [167,168]. Extracellular NMN may have to be first converted into NR by the enzyme CD73 before it can be transported into the cells. Once it has been absorbed into the liver, NA enters the Preiss-Handler pathway to form nicotinic acid mononucleotide (NAMN), which is mediated by nicotinic acid phosphoribosyl transferase (NAPRT). After another conversion into nicotinic acid adenine dinucleotide (NAAD), ultimately, NAD is formed in this pathway by NAD synthase (NADS). NAM, on the other hand, is converted by nicotinamide phosphoribosyl transferase (NAMPT) into NMN, which is further processed to become NAD by NMN adenylyl transferases (NMNAT1-3) in the NAD salvage pathway. NR is also metabolized in the salvage pathway by forming NMN after phosphorylation catalyzed by the enzymes NRK1 and NRK2 [169]. NRK1 and NRK2 also catalyze the phosphorylation of nicotinic acid riboside (NAR) to form nicotinic acid mononucleotide (NAMN) in the Preiss-Handler pathway. NAR is another bioavailable NAD precursor that is also produced in human cells [165,170].

NAD itself has also been investigated to increase the levels of NAD for the treatment of schizophrenia and PD, with varying success in the clinical trials [171,172,173]. This strategy to elevate body-wide NAD levels appears to be counterintuitive at first because NAD, such as other nucleotides, does not readily cross cell membranes. While the process of NAD absorption into cells is not yet well understood, it has been shown that circulating NAD can be converted into NAM in the extracellular space through the NAD hydrolase activity of the membrane-bound ecto-enzymes CD38 and CD157 (Figure 1). NAD may also be directly converted into NMN by CD73. Dietary supplementation with tryptophan can augment NAD synthesis via the kynurenine (de novo) pathway, which ultimately produces quinolinic acid (quinolinate), which is converted by QPRT into NAMN, which enters the Preiss-Handler pathway. A comprehensive review providing detailed information on the doses and results of the clinical studies investigating the supplementation of NAD precursors has been previously published [161].

While all of the supplements ultimately aim to increase the NAD levels, it appears that different precursors are, surprisingly, not entirely redundant, and different precursors might work better for certain cell or tissue types [174]. NAD deficiency has been observed in numerous diseases unrelated to aging-related NAD decline, such as degenerative diseases [175], hearing loss [176,177], skeletal muscle dysregulation [175,178], and mitochondrial pathologies [179], among others. The beneficial effects of NR supplementation to bolster the NAD concentrations have also been reported in a zebrafish model of muscular dystrophy [180]. Similarly, mice with mitochondrial myopathy and muscular dystrophy treated with NR demonstrated a delayed disease progression, likely through enhanced mitochondrial biogenesis and delayed stem cell senescence [179,181]. NAD-deficient human patients with mitochondrial myopathy, which developed when they were adults, were given a niacin supplementation, and there was an increase in their muscle NAD concentrations, muscle strength, and mitochondrial function [175]. Additionally, mouse models for Cockayne syndrome (CS), a progressive aging disease characterized by premature hearing loss, resulted in reduced cochlear NAD concentrations [176,177]. An NR treatment was reported to restore outer hair cell loss while aiding in the prevention of hearing loss in CS mouse models. A niacin treatment has also been shown to reduce the amount of plasma triglycerides and hepatic fatty acid synthesis in rabbits, while also causing the regression of fatty liver disease in rats [182,183]. Recently, however, Williams et al. examined the effects of NR in obese mice and concluded there was little evidence to support that supplementation increases the NAD levels in the tissues or improves the mitochondrial and skeletal muscle functions [184], suggesting that more work needs to be completed in this area.

Oral NA supplementation prevents photo-carcinogenesis and photo-immunosuppression in mice [185], however, the effectiveness of oral vitamin B3 supplementation in the treatment of skin aging may be limited due to the substantial biotransformation of these NAD precursors in the liver. The pioneering work in Myron and Elaine Jacobson’s laboratories showed that the topical application of myristylated, and hence, highly lipophilic, NA formulations can be used to increase the skin barrier’s function, epidermal differentiation, and the skin NAD content to prevent or ameliorate retinoic acid-induced epidermal thinning [186,187].

In addition to the inhibition of NAD-consuming enzymes and supplementation with dietary precursors, the pharmacological inhibition of ACMSD has been proposed as a third strategy to elevate the NAD levels in the liver and kidneys [188,189]. In the de novo NAD synthesis pathway (Figure 2), the unstable metabolic intermediate ACMS is spontaneously cyclized to form quinolinate, which is converted into NAMN by QPRT. The liver- and kidney-specific enzyme ACMSD limits the availability of ACMS for non-enzymatic conversion into quinolinate by converting ACMS to picolinate, which is ultimately converted into acetyl-CoA (Figure 1). The inhibition of ACMSD therefore increases the NAD levels, which may protect the liver and kidney tissues from NAD-depletion associated injury. It was also discovered that the FDA-approved non-steroidal anti-inflammatory drug diflunisal and some of its derivatives are competitive inhibitors of ACMSD, causing these drugs to modulate the NAD levels in short-term applications [190].

Finally, the NAD levels may be increased in the body by decreasing the elimination and excretion of NAM by inhibiting its conversion into mNAM, which is catalyzed by the enzyme NAM N-methyltransferase (NNMT) (Figure 1). NNMT has a significant impact on NAD homeostasis since mNAM cannot be converted into NAD, therefore, NNMT activity directly reduces the amount of free NAM available for NAD synthesis in the salvage pathway. By regulating the NAM levels, NNMT also exerts a regulatory effect on SIRT1 because NAM is a potent inhibitor of SIRT1, while mNAM was also shown to stabilize SIRT1 protein, thereby enhancing SIRT1-dependent gene activity [191,192]. In line with these findings, the overexpression of an NNMT isoform to increase the mNAM levels has been shown to increase the lifespan of C. elegans by stabilizing SIRT1 [193]. Interestingly, a genetic knock-down of NNMT expression in white adipose tissue and the liver protected the mice from high-fat-diet-induced obesity by augmenting cellular energy expenditure, further making NNMT an interesting drug target [194]. Selective NNMT inhibitors have been developed that could be used for treatment of several pathological conditions including cancer, obesity, metabolic disorders, and alcohol-related fatty liver disease [195,196,197,198].

## 6. Summary/Conclusions

The insight that low NAD levels are causally involved in many aging-related and metabolic disorders has led to renewed interest in NAD-related basic and translational research [199]. Modern NAD research today has achieved far more than it did in the beginning when pellagra was still a significant health problem, and it now addresses some of the most pressing health issues of our time, such as aging-related pathologies, neurodegenerative conditions, and metabolic disease. Dietary and pharmacological supplementation strategies aimed at maintaining or restoring healthy NAD metabolite levels are currently subject to intense research. As we are still exploring the mechanistic and molecular effects of suboptimal NAD levels, it remains essential to develop and use laboratory animal models that enable the investigation of moderate, but chronic NAD deficiency and gradual NAD decline, which are thought to be at the root of human NAD-related pathologies. Such animal models will serve as valuable tools in the development of appropriate intervention strategies, but as it is laid out in this review, a range of different models will be needed to investigate the NAD metabolic pathways that have been emerging as compartmentalized, tissue specific, and more complex than they were anticipated to be.

## Figures and Tables

**Figure 1 cells-12-00500-f001:**
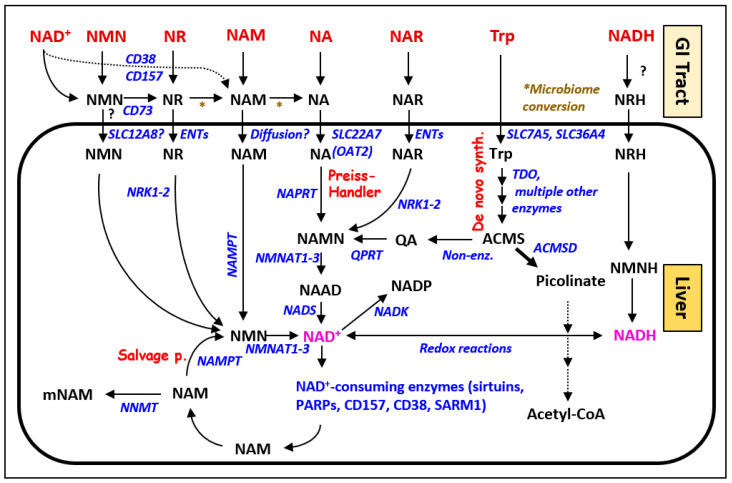
Precursor molecules used for boosting NAD metabolism. Orally administered compounds (top row, red font) are metabolized by the gut microbiome to four main bioavailable metabolites: nicotinamide riboside (NR), nicotinamide (NAM), nicotinic acid (NA), and nicotinic acid riboside (NAR), which enter cells either by passive diffusion or with the help of cellular transporters, as indicated (* symbol designates microbiome conversion reactions). Extracellular nicotinamide adenine dinucleotide (NAD) and nicotinamide mononucleotide (NMN) are metabolized by extracellular NAD hydrolases CD38/CD157 and ecto-nucleotidase CD73, respectively, to NR, which crosses hepatic cell membranes with the help of equilibrative nucleoside transporters (ENTs). Once they are inside the (hepatic) cell, NR and NAM are converted into NMN by nicotinamide riboside kinases NRK1-2 and nicotinamide phosphoribosyltransferase (NAMPT), respectively, and enter the salvage pathway. In the cell, NA and NAR are metabolized into nicotinic acid mononucleotide (NAMN) by the enzymes nicotinic acid ribosyltransferase (NAPRT) and NRK1-2, respectively, to enter the Preiss-Handler pathway. Dietary tryptophan (Trp) is transported into the cell by several transporters, including SLC7A5 and SLC36A4, where it is metabolized by TDO and IDO to enter the kynurenine (de novo) NAD synthesis pathway (see Figure 2) to form NAMN, which enters the Preiss-Handler pathway to form NAD. The uptake and metabolism of NADH are poorly understood, but they likely include extracellular conversion steps. The different fates of NAD in the cell include phosphorylation by NAD kinase (NADK) to NADP^+^/NADPH and cleavage by NAD consuming enzymes into ADP-ribosyl moieties transferred to target molecules or to cyclo-ADP-ribose (not shown here) and NAM. NAM is then recycled in the salvage pathway, where it is converted into NMN by NAMPT. The clearance of NAM is achieved by S-adenosyl methionine-dependent methylation, which changes it into N-methylnicotinamide (mNAM), a reaction mediated by nicotinamide N-methyltransferase (NNMT).

## Data Availability

Not applicable.

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
