# Peer review of "Beyond Pellagra—Research Models and Strategies Addressing the Enduring Clinical Relevance of NAD Deficiency in Aging and Disease"

_cells, 2023, doi:10.3390/cells12030500_

Round 1
Reviewer 1 Report
This review paper by Feuz and colleagues explores the problem of NAD deficiency in modern times, where pellagra is very infrequently diagnosed, but where a decline in NAD levels is receiving increasing attention because of its implications in aging.
The work is interesting and easy to read. I would suggest to change the title: the word "issues" is very vague and perhaps the term "aging" should appear in the title, as it is a leit motif of the review. Also, the authors could think about re-organizing their work, including the data coming from different animal and cellular models in the first part of the review concerning "NAD deficiency problem". In my view, this re-organization of data presentation could present a more cohesive case for the last paragraph on strategies to increase NAD levels.
Lastly, the figures are quite complex. Figure 2 could be made smaller, but should include the phenotype of the animals, otherwise it is simply a list of models, without functional implications.
Author Response
Dear Reviewer: Thank you for your comments, which have been very helpful for improvement of the manuscript.
- We changed the title as you suggested, which is now much more comprehensive and representative of the review's contents. It is now: 'Beyond Pellagra – Research Models and Strategies Addressing the Enduring Clinical Relevance of NAD Deficiency in Aging and Disease'.
- Figure 2 was updated to include a brief description of the mouse model and its main phenotype.
- We tried to reorganize the paper as you suggested. We found that a surprisingly large portion of the available data in the literature was actually not initially produced in animal models like in most other research fields. Organizing the manuscript in a way that presents data obtained from cells and animals first and then move on to the other parts made it very difficult to maintain logical flow of the review. After much debate, we unfortunately had to decide to keep the current structure of the review for readability.
Reviewer 2 Report
The review “Beyond Pellagra – Ongoing Issues and Models of Niacin and NAD Deficiency” is a manuscript regarding the actual state-of-art and ongoing issues on pathological conditions associated to NAD deficiency. The manuscript is well organized and comprehensively described. However, there are some concerns that authors should address in order to improve the work:
1. A paragraph regarding the role of nicotinamide N-methyltransferase (NNMT) in NAD homeostasis is mandatory. Indeed, by methylating nicotinamide, NNMT can regulate the NAD levels reducing the amount of free nicotinamide which could be converted into NAD through the NAD-salvage pathway. Nevertheless, NNMT has been demonstrated in several studies to exert a regulatory effect on SIRT1 through the regulation of its substrate, the nicotinamide, which is a potent inhibitor of SIRT1, while methylnicotinamide was demonstrated to stabilize the SIRT1 activity (PMID: 34153425; PMID: 26168293). While methylnicotinamide was itself proven to increase lifespan (PMID: 24077178), many NNMT inhibitors have been developed which could be used the treatment of several pathological conditions including cancer, obesity, metabolic disorders, alcohol-related fatty liver etc.. (PMID: 34704059; PMID: 34572571; PMID: 29483571; PMID: 29155147; PMID: 34424711).
Thus, the complex interaction of these elements must be taken into account since it has a great impact on NAD homeostasis.
2. A table summarizing the studies together with the concentrations and models involved regarding NAD precursors supplementation would be beneficial for the manuscript.
Author Response
Dear Reviewer: Thank you for your helpful and insightful comments, which we have taken into full consideration for improvement of the manuscript.
- A paragraph detailing the role of NNMT in NAM metabolism has been added to the manuscript, as suggested, which we believe nicely complemented the paper (lines 612-626).
- An excellent comprehensive review by Radenkovic, Reason and Verdin (2020) of clinical studies with concentrations and models regarding NAD precursors supplementation has already been published, which we cite in lieu of duplicating this effort (lines 568-570).
Reviewer 3 Report
Overall, the manuscript was well-written, providing comprehensive and current knowledge about NAD deficiency related issues and research models to study NAD deficiency. Minor revision is needed to address the following:
1) NADH/H+ does not appear to be a common way for the reduced form of NAD. It may be simpler just to use NADH.
2) Line 81-83. “The recent years have seen accumulating evidence that decreasing NAD levels are associated with multiple disease states and health problems, ranging from metabolic dysfunction, neurodegeneration, and inflammatory diseases to problems with fertility and embryo development”. It is better to provide a citation or citation(s).
3) If possible, please add in some contents about Nampt KO animal models (such as tissue-specific Nampt KO models) and how age-associated NAD relates to NAD biosynthesis in the salvage pathway.
4) Line 537 “produced human cells”, it seems “in” is missing after “produced”?
5) Line 566-568, please specify that Williams’ examination was performed in obese mice.
Author Response
Dear Reviewer: Thank you for your kind comments and suggestions, which were implemented in their entirety in the revised manuscript:
- The designation NAD/H+ was changed to NADH throughout the paper
- References were added to support the sentence in lines 84-87, as suggested.
- A paragraph reviewing Nampt ko mice was added, including the appropriate references (lines 475-488). In addition, Figure 2 was updated accordingly, including the figure legend. A complementary sentence mentioning mouse models that do not show lower NAD levels was included in lines 489-93. In addition, two sentences clarifying how age-associated NAD relates to NAD biosynthesis in the salvage pathway were added to section 2.1 NAD and Aging (lines 104-107).
- The sentence in line 562 was corrected to "NAR is another bioavailable NAD precursor that is also produced in human cells [160,165]."
- A sentence was clarified to read "Recently, however, Williams et al., examined the effects of NR in obese mice and concluded there was little evidence to support that supplementation increases NAD levels in the tissues or improves mitochondrial and skeletal muscle function [179], suggesting more work needs to be completed in this area."